# Secrecy Control of Wireless Networks with Finite Encoding Blocklength

**Qiuming Liu, Shumin Liu, Chunshui Zeng, Xiaohong Qiu and He Xiao ***

Nanchang of Jiangxi University of Science and Technology, 1180 Shuanggang Road, Nanchang 330013, China; liuqiuming@hust.edu.cn (Q.L.); liushumin@jxust.edu.cn (S.L.); zengchunshui@jxust.edu.cn (C.Z.); qiuxiaohong.jxust@gmail.com (X.Q.)

*   Correspondence: xiaohe804@gmail.com; Tel.: +86-791-83811871; Fax: +86-791-83811871

**Abstract:** We consider wireless multi-hop networks in which each node aims to securely transmit a message. To guarantee the secure transmission, we employ an independent randomization encoding strategy to encode the confidential message. We aim to maximize the network utility. Based on the finite length of a secrecy codewords strategy, we develop an improved control algorithm, subject to network stability and secrecy outage requirements. On the basis of the Lyapunov optimization method, we design an control algorithm, which is decomposed into end-to-end secrecy encoding, flow control and routing scheduling. The simulation results show that the proposed algorithm can achieve a utility result that is arbitrarily close to the optimal value. Finally, the performance of the proposed control policy is validated with various network conditions.

**Keywords:** wireless networks; secrecy encoding; network control; Lyapunov optimization

## 1. Introduction

A wireless multi-hops network is a decentralized network which is contained by a set of nodes. The message is transmitted over the wireless channel with multi-hops mode. Since the wireless channel is a broadcast channel, it is easy to eavesdrop on the data transmission. Therefore, when designing a network, in addition to considering the quality of service (QoS) constraint, the data transmission security should also be involved. In 1975, Wyner [1] studied a wire-tapped noisy channel and encoded the data to confuse the eavesdropper. After ground-breaking work in [1], many works have tackled the secrecy transmission of wireless networks. Liang et al. [2] considered a cellular network where the base station needed to transmit data to multiuser confidentially, they designed a dynamic control algorithm to maximize the network utility by employing infinite secrecy encoding block. Since secrecy outage was inevitable, Wang et al. [3] investigated the secrecy outage and and secrecy rate in a multiuser wireless systems. Jointly considering the reliability, security and stability, they designed an optimal online control algorithm by exploiting stochastic network optimization method. Later, in [4], the authors also considered a multi-user wireless scenario with imperfectly known channels. They aimed to maximize the worst case of secrecy throughput and developed a low complexity and rapid convergence algorithm for the optimal power allocation. In [5], Koksal et al. investigated a cross-layer security in wireless network and proposed a dynamic control algorithm to maximize the network utility. After that, in [6], aiming to minimize the secrecy outage probability, the authors extended the dynamic control algorithm to a cognitive radio network and developed a bandwidth and resource allocation algorithm.

Recently, on the basis of the works mentioned above, the secrecy control problem in wireless networks has been extensively studied, such as the OFDMA-based wireless network and cognitive wireless network. In [7,8], the authors formulated an analytical framework for secure resource

allocation in a downlink OFDMA-based broadband network. Jointly considering the power and subcarrier allocation, they proposed an optimal algorithm to maximize the average aggregate rate of all users for a base station. In [9], the authors extended the cellular network to cognitive network and proposed a scheduling policy to maximize the secrecy rate of second users. In [10,11], they designed a scheduling policy in cognitive radio networks to analyze the ergodic capacity on the impact of fading channel and distributed eavesdroppers. In [12], Maged et al. considered the cognitive wiretap channel and proposed multiple antennas to secure the transmission at the physical layer. They revealed the impact of the primary network on the secondary network in the presence of a multi-antenna wiretap channel.

Although these security control algorithms improved the network performance, most of them focused on the cellular networks. In contrast, for the multi-hop wireless networks, there were few works involved. In [13,14], the authors considered a multi-hop wireless network and proposed an optimal control to maximize the network throughput. However, they did not consider the data transmission security. In [15,16], the authors investigate the confidential message secure transmission in a large scale wireless networks. By using infinite secure encoding blocklength, they derived the secrecy throughput with ideal control policies. After that, Zheng et al. [17] extended the infinite secure encoding blocklength to the finite case and revealed the relation of secrecy outage probability to the length of codewords. As for the mobile ad hoc networks, Li et al. [18] jointly exploited cooperative jamming and secrecy guard zone scheme and derived the exact secrecy throughput based on the physical layer security technology. To enhance security, Zhu et al. [19] employed directional antennas and evaluated the secure secrecy performance in millimeter wave ad hoc networks. While for the wireless multi-hop networks, He [20] proposed a dynamic control algorithm in multi-hop wireless network with untrusted relays. While in [21], Sarikaya et al. considered a multi-hop network with random and independent node distribution. To guarantee the secrecy transmission, they developed a strategy to encode the confidential message with multi-path transmission and infinite coding block length. By using stochastic network optimization [22], they developed a control policy to stable the network and maximize the network utility, which combines end-to-end secrecy encoding, routing scheme and the resource allocation algorithm. However, most of the works mentioned above only focused on cellular networks security or multi-hop wireless networks with multi-path transmission strategy to guarantee the security.

In this paper, we consider the secrecy control problem in multi-hop wireless network, which is extended work of our previous work [23]. In [23], we have developed a secrecy control algorithm to maximize the network utility, while the blocklength of secrecy encoding is infinite. For the case of finite blocklength, it would be much more complicated since perfect secrecy is not possible. To deal with this scenario, We exploit an independent randomization encoding strategy to guarantee the security and define a secrecy outage probability. Given a constraint on the probability of secrecy outage, we develop an improved control algorithm, which is decomposed into end-to-end secrecy encoding, flow control and routing scheduling such that the network stability and secrecy outage constraint are satisfied. Finally, we prove that the performance of proposed control policies can close to the optimal utility result asymptotically.

The rest of the paper is organized as follows. In Section 2, we introduce the network model and problem formulation. Section 3 proposes an improve control policy with finite secrecy codewords. Section 4 evaluates the proposed policy with various network conditions. Finally, the paper is concluded in Section 5.

## 2. System Models

### 2.1. Network Model

The wireless Ad hoc network is formed by $M$ legitimate nodes and $L$ links connecting the nodes. As shown in Figure 1, for a link $l \in \{1, 2, ..., L\}$, let $T(l)$ and $D(l)$ be the set of transmitter and receiver

nodes on link $l$. The eavesdroppers set is denoted as $E$. In the network system, each node wishes to transmit its confidential message to the destination via a multi-hops manner against eavesdroppers. We assume the network operates on a time-slotted model and the slot is normalized to integral unit $t \in \{0, 1, 2, ...\}$. In this work, there exists a reasonable assumption that the system Channel State Information (CSI) is known. As in [10], each node can get full-CSI by utilizing pilot symbols and CSI feedback process. For example, each node reports a received-signal-strength index to PBS in packets such as RSSI reports. Let $\vec{S}(t) = (S_1(t), ..., S_L(t))$ represent the channel state vector of link set $L$ in slot $t$, which is a block fading channel and follows independent and identically distribution (i.i.d) and $S_l(t)$ is the channel state of link $l$. Note that $S_l(t)$ contains $N < \infty$ channels which implicates the perfect secrecy [1] cannot satisfy. Let $R_l(t)$ and $\bar{R}_l^e(t)$ denote the achievable rate on link $l$ and the maximum overhearing rate of eavesdropper $e$, respectively. Since the network is a multi-commodity problem, each flow is identified by its destination node $c \in \{1, ..., M\}$. Let $\lambda_n^{cs}(t)$ be the arrival confidential data at node $n$ and destined for node $c$, which is bounded by $\lambda_n^{c\max}$. As shown in Figure 1, there is a control valve at each node to admit $R_n^{cs}(t)$ confidential data into the network. For each flow $c$, let $Q_n^c(t)$ be the queue backlog at node $n$. If $c = n$, $Q_n^n(t) = 0$ for all $n$ and $t$.

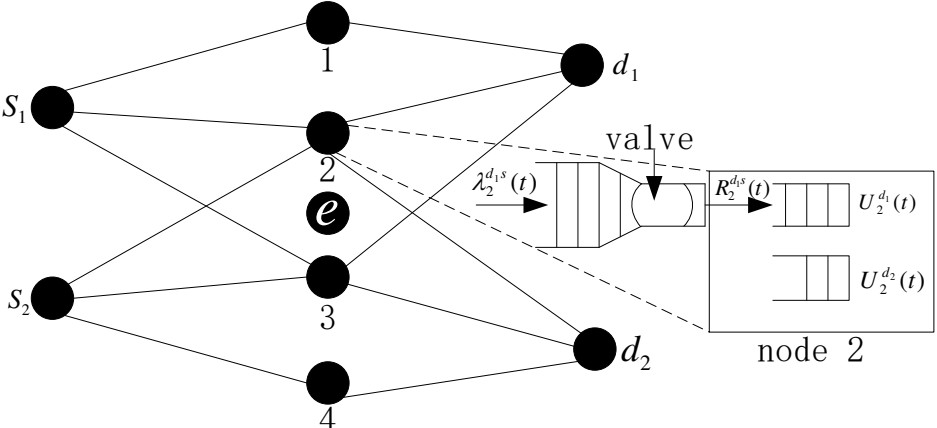

**Figure 1.** Network model.

Since the data is transmitted by wireless channel, there exists simultaneous transmission interference. Thus, Link set $L$ can not be fully utilized. Let $g$ be a link set that can be transmitted simultaneously, and $G$ denote the collection of all sets of the link set $g$, where set $G$ is determined by the network interference model. We also define an indicator variable $I_g(t)$ to represent the decision of the network in slot $t$, where $I_g(t) = 1$ means the link set $g$ is scheduled in slot $t$, otherwise it is 0. Similarly, indicator variable $I_{ij}^{n,c} = 1$, if link $(i,j)$ is employed to transmit the flow generated by source $n$ to node $c$, otherwise it is 0. Thus, in slot $t$, the flow rate of source $n$ to node $c$ at link $(i,j)$ is

$$\mu_{ij}^n(t) = \begin{cases} R_{ij}(t) & \text{if } (i,j) \in g, I_g(t) = 1, I_{ij}^{n,c} = 1. \\ 0 & \text{else} \end{cases} \tag{1}$$

and the time average link rate is

$$\bar{\mu}_{ij}^n = \lim_{t \to \infty} \frac{1}{t} \sum_{\tau=1}^{t} \mu_{ij}^n(\tau). \tag{2}$$

Due to the broadcast feature of wireless channel, the confidential message is overheard by eavesdroppers. To guarantee the secure transmission of confidential message, we employ Wyner's encoding scheme [1] to encode the confidential message. Specifically, at each slot $t$ and link $l$, using independent randomization encoding strategy, the transmitter, according to $R_l(t)$ and $\bar{R}_l^e(t)$, encodes a mount of $R_n^s(t)$ (may contain multiple flows) confidential message from its arrival data. Such that

the maximum output confidential message rate $R_l^s(t)$ can be denoted as $R_l(t) - \bar{R}_l^e(t)$, if $R_l^s(t) > R_l(t) - \bar{R}_l^e(t)$, secrecy outage occurs. Using this secrecy encoding strategy, we can guarantee the transmission security in each link. All the parameters that have been defined or would be used are presented in Table 1.

**Table 1.** Notations.

| Notation | Definition |
|----------|------------|
| $M$ | Number of legitimate nodes. |
| $E$ | The set of eavesdroppers. |
| $N$ | Number of channels. |
| $T(l), D(l)$ | Transmitter and receiver nodes of link $l$. |
| $S_l(t)$ | Channel state of link $l$ in slot $t$. |
| $R_l(t)$ | Achievable rate on link $l$ in slot $t$. |
| $\bar{R}_l^e(t)$ | Maximum leakage rate at eavesdropper $e$. |
| $\lambda_n^{cs}(t)$ | Confidential data arrival rate. |
| $\lambda_n^{c\,\max}(t)$ | The maximum Confidential data arrival rate. |
| $R_n^{cs}(t)$ | Confidential data admitted into the network. |
| $R_l^s(t)$ | Secrecy encoding rate. |
| $\mathbf{Q}(t), \mathbf{P}(t), \mathbf{Z}(t), \mathbf{Y}(t)$ | Queues backlog in the network. |
| $V$ | Parameter of control algorithm. |
| $U_n^c(\cdot)$ | Utility functions. |

## 2.2. Problem Formulation

In [23], we have proposed a control algorithm according to the secrecy encoding, where the confidential message is encoded into an infinite codewords. Thus, it would involve an infinitely long delay to decode the confidential message. In this paper, we remove the assumption of infinite codewords, i.e., $N < \infty$. Since the confidential message is encoded into a finite codeword, the perfect secrecy for all message can not be guaranteed. Thus, to embody the security of confidential message, we define the notion of secrecy outage. The secrecy outage occurs if the confidential message is intercepted by eavesdroppers. To evaluate the state of secrecy outage, we assume each source node can collect the knowledge of the confidential message accumulated by eavesdroppers. Such that the source node can identify the occurrence of secrecy outage. Although this assumption is somewhat ideal, it provides a better insight on the performance of secure communication in multi-hop network. In addition, according to the state of secrecy outage, each secrecy codeword $k$ would encode different confidential message $R_n^{k,cs}$. Thus, an encoding scheme needs to be designed to satisfy the requirement of secrecy outage. Let $R_n^{p,cs}$ be the average confidential message rate transmitted from node $n$ to $c$, $p_n^{out}(R_n^{k,cs})$ denote the average secrecy outage of codeword $k$ at node $n$, $\gamma_n$ be the maximum allowable portion of confidential message intercepted by the eavesdropper.

We aim to determine a joint scheduling, secrecy encoding scheme and routing algorithm that maximizes aggregate network utility. Let $U_n^c(x)$ be utility obtained by source $n$ destined to node $c$ when the confidential transmission rate is $x$ bits/channel use. We assume that $U_n^c(\cdot)$ is a continuously differentiable, increasing and strictly concave function. There is a finite backlog at the transport layer, which contains the secrecy-encoded messages. In each slot, source node $n$ determines the amount of encoded information admitted to its queue at the network level. Let $\lambda_n^{cs}(t)$ be the amount of traffic injected into the queue of source $n$ and destined to node $c$ at slot $t$. Our objective is to support the traffic demand to achieve a long term confidential rate that maximizes the sum of utilities. Then the optimization problem can be formulated as following:

$$\max_{R_n^{p,cs}(t), I_g(t), I_{ij}^{n,c}(t)} : \quad \sum_{n,c} U_n^c(\bar{r}_n^{p,cs}) \tag{3}$$

$$\text{s.t.} \quad 0 \le R_n^{p,cs}(t) \le \lambda_n^{cs}(t) \text{ for all } (n,c) \tag{4}$$

$$\sum_{\{j|(i,j)\in L\}} \bar{\mu}_{ij}^n - \sum_{\{i|(i,j)\in L\}} \bar{\mu}_{ij}^n \geq 0 \tag{5}$$

$$\bar{R}_n^{out} \leq \gamma_n \bar{R}_n^{cs} \tag{6}$$

where $\bar{R}_n^{cs} = \lim_{K\to\infty} \frac{1}{K} \sum_{k=1}^K R_n^{n,cs}$, $\bar{R}_n^{out} = \lim_{K\to\infty} \frac{1}{K} \sum_{k=1}^K R_n^{n,cs} p_n^{out}(R_n^{k,cs})$. Constraint (4) guarantees the average confidential message rate is not larger than the message arrival rate; Constraint (5) is the input flows and output flows constraint at the intermediate nodes; Constraint (6) is the requirement of maximum allowable portion of confidential message intercepted by eavesdropper.

## 3. Control with Finite Secrecy Codewords

Similar to the control algorithm proposed in [23], we exploit the Lyapunov penalty and drift to solve this problem. However, due to the secrecy codeword finite and secrecy outage occurrence, the queue model needs to be improved. In particular, as shown in Figure 2, source node is equipped with two separate queues which are operated at two different time scales. The first queue stores the message admitted into the network and $Q_n^{p,c}(t)$ denotes the queue length in slot $t$. Let $R_n^{p,cs}(t)$ be the admitted confidential message in slot $t$, which is transmitted from source node $n$ to destination node $c$. $\bar{R}_n^{p,cs}$ represents the long-term average admitted confidential message. The departure of the first queue occurs only when a new secrecy codeword is generated in slot $t$. Let $k_n(t)$ be the number of secrecy codewords generated in slot $t$, and $R_n^{k_n(t),cs}$ denote the confidential message encoded in $k_n(t)$-th secrecy codeword. Since there are $N$ channels, where $N < \infty$, the actual transmitted confidential message is $NR_n^{k_n(t),cs}$ and the length of secrecy codeword is $NR_n^c$, the second queue is a partial queue and let $P_n^c(t)$ denote the queue length. In this queue, the data departures or not is depended the scheduling and routing policy. Only when the queue is empty, i.e., $P_n^c(t) = 0$, a new secrecy codeword is allowed to admit into the queue. Thus, we have, if $P_n^c(t) = 0$, $k_n(t+1) = k_n(t) + 1$. According to the queue models defined above, the evolution of queues can be expressed as:

$$Q_n^{p,c}(t+1) = \begin{cases} \left[Q_n^{p,c}(t) - NR_n^{k_n(t+1),cs}\right]^+ + R_n^{p,cs}(t) & \text{if } P_n^c(t) = 0. \\ Q_n^{p,c}(t) + R_n^{p,cs}(t) & \text{else} \end{cases} \tag{7}$$

$$P_n^c(t+1) = \begin{cases} NR_n^c & \text{if } P_n^c(t) = 0. \\ \left[P_n^c(t) - \sum_{i|(n,i)\in L} \mu_{ni}^n(t)\right]^+ & \text{else} \end{cases} \tag{8}$$

For each intermediate node, there exists a queue to store the packet from source node $n$ to destination node $c$. Let $Q_i^{n,c}(t)$ be the queue length. Then we have

$$Q_i^{n,c}(t+1) = \left[Q_i^{n,c}(t) - \sum_{j|(i,j)\in L} \mu_{ij}^n(t)\right]^+ + \sum_{j|(j,i)\in L} \mu_{ji}^n(t). \tag{9}$$

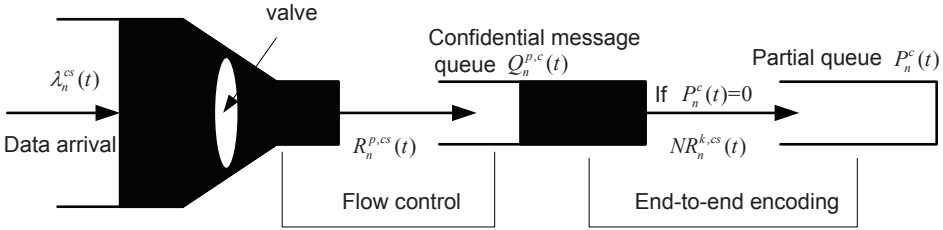

**Figure 2.** Queues of source nodes for finite secrecy codewords.

To evaluate the state of secrecy outage, we assume source node $n$ can perceive the accumulated of confidential message at each eavesdropper, since it can check if some eavesdropper has intercepted

the transmission. If it is impossible to know the information precisely, we will discuss in Section 4. Let $Z_e^{n,c}(t)$ denote the number of bits that must be accumulated by eavesdropper $e$ in slot $t$ to decode the $k_n(t)$-th confidential message. Note that, for any eavesdroppers $e$, if $Z_e^{n,c}(t) = 0$, then secrecy outage of the $k_n(t)$-th confidential message is occurred. Let $\bar{R}^e(t)$ be the maximum rate that eavesdropper can be achieved, then the evolution of queue $Z_e^{n,c}(t)$ can be denoted as

$$Z_e^{n,c}(t+1) = \begin{cases} \left( R_n^c - R_n^{k_n(t+1),cs} \right) N & \text{if } P_n^c(t) = 0. \\ \left[ Z_e^{n,c}(t) - \bar{R}^e(t) \right]^+ & \text{else} \end{cases} \tag{10}$$

For the constraint (6), a virtual queue $Y_n^{k,c}$ is constructed, which is employed to record the state of secrecy outage. Thus, if the $k$-th confidential message is secrecy outage, then the arrival rate of queue $Y_n^{k,c}$ is $R_n^{k,cs}$, else it is 0. The evolution of queue $Y_n^{k,c}$ is:

$$Y_n^{k+1,c}(t+1) = \begin{cases} \left[ Y_n^{k,c}(t) + R_n^{k,cs} - \gamma_n R_n^{k,cs} \right]^+ & \text{if k-th message is outage.} \\ \left[ Y_n^{k,c}(t) - \gamma_n R_n^{k,cs} \right]^+ & \text{else} \end{cases} \tag{11}$$

The arrival and departure of virtual queue $Y_n^{k,c}$ is the number of confidential message bits undergoing secrecy outage, and the number of confidential message bits allowed to be outage, which is constrained by parameter $\gamma_n$. The value of $Y_n^{k,c}$ indicates the amount that we have exceeded the allowable secrecy outage constraint. Hence, the larger the value of $Y_n^{k,c}$, the more conservative our control algorithm needs to be to meet these constraints. That is, a less confidential message $R_n^{k_n(t),cs}$ is encoded into the $k_n(t)$-th codeword.

It is necessary to introduce the concept of strong stability. As a discrete time process, $Q(t+1) = [Q(t) - \mu(t)]^+ + \lambda(t)$ is strongly stable if:

$$\limsup_{t \to \infty} \frac{1}{t} \sum_{\tau=0}^{t-1} \mathbb{E}\{Q(\tau)\} < \infty, \tag{12}$$

In particular, a multi-queue network is stable when all queues of the network are strongly stable. According to Strong Stability Theorem in [13], for finite variable $\mu(t)$ and $\lambda(t)$, strong stability implies a rate stability of $Q(t)$. The definition of rate stability can be found in [13] and omitted here.

### 3.1. Control Algorithm

For the secrecy transmission strategy with finite secrecy codewords, the control algorithm is:

**Multi-hop Secrecy Control Algorithm with Finite Block (MSCAFB):**

- **End-to-End Encoding**: At each new secrecy codeword generation, i.e., $P_n^c(t) = 0$, let $k_n(t+1) = k_n(t) + 1$, the confidential message $R_n^{k_n(t+1),cs}$ allowed to encode into a new codeword satisfies:

$$R_n^{k_n(t+1),cs} = \arg\max_r \left\{ r \left[ Q_n^{p,c}(t) - Y_n^{k_n(t),c}(p_n^{out}(r) - \gamma_n) \right] \right\}. \tag{13}$$

- **Flow Control**: In each slot $t$, for a given parameter $V$, the admitted confidential message at each source node $n$ is:

$$R_n^{p,cs} = \arg\max_x \left\{ V U_n^c(x) - x Q_n^{p,c}(t) \right\}, \tag{14}$$

- **Scheduling**: In each slot $t$, if $I_g(t) = 1$ and $I_{ij}^{n,c}(t) = 1$, then the flow of node $n$ is on the link $(i, j) \in g$ and the scheduler selects the set of $l^*$, where

$$
\begin{aligned}
(n, l^*) = \arg\max_{n,g} \Bigg\{ & \sum_{(n,i)\in g} \mu_{ni}^n(t) \left( \frac{R_n^c}{R_n^{k_n(t),cs}} Q_n^{p,c}(t) + P_n^c(t) \right. \\
& \left. - Q_i^{n,c}(t) \right) + \sum_{(i,j)\in g} \mu_{ij}^n(t) \left( Q_i^{n,c}(t) - Q_j^{n,c}(t) \right) \\
& - \sum_{e\in E} \sum_{e\neq i} Z_e^{n,c}(t) \bar{R}^e(t) \Bigg\}.
\end{aligned}
\tag{15}
$$

The term $\frac{R_n^c}{R_n^{k_n(t),cs}}$ of $Q_n^{p,c}(t)$ is used to normalize it to the value of other queues.

Note that, the long-term average secrecy outage $p_n^{out}(r)$ is increasing with variable $r$. Once $r$ increases, secrecy codeword is encoded with less randomization bits, such that eavesdropper can intercept the confidential message with a higher probability. Hence, as the queue length $Q_n^{p,c}(t)$ increases, the confidential message $R_n^{k_n(t),cs}$ is increased. Moreover, it decreases with the increasing of the virtual queue length $Y_n^{k_n(t),c}$, such that the constraints of problem (3) are satisfied.

*3.2. Algorithm Performance*

Using the Theorem of Lyapunov penalty and drift [22], we can also prove that the proposed control algorithm can close to the optimal arbitrarily. Let $\mathbf{Q}^p(t) = (Q_1^{p,c}(t), \ldots, Q_n^{p,c}(t))$ denote the queue vector of the first queue in Figure 2, $\mathbf{P}_n(t) = (P_1^c(t), \ldots, P_n^c(t))$ be the queue vector of the second queue in Figure 2, $\mathbf{Q}^n(t) = (Q_1^{n,c}(t), \ldots, Q_i^{n,c}(t))$ represent the queue vector of intermediate nodes for flow $c$, $\mathbf{Z}^n(t) = (Z_1^{n,c}(t), \ldots, Z_e^{n,c}(t))$ be the queue vector of leaked confidential message accumulated by eavesdropper and $\mathbf{Y}_n^k = (Y_1^{k,c}, \ldots, Y_n^{k,c}(t))$ be the queue vector of secrecy outage. All the queue vectors can be denoted as $\mathbf{\Theta}(t) \triangleq \left( \mathbf{Q}^p(t), \mathbf{P}_n(t), \mathbf{Q}^n(t), \mathbf{Z}^n(t), \mathbf{Y}_n^k \right)$. Define a Lyapunov function as

$$
\begin{aligned}
L(\mathbf{\Theta}(t)) &\triangleq L \left( \mathbf{Q}^p(t), \mathbf{P}_n(t), \mathbf{Q}^n(t), \mathbf{Z}^n(t), \mathbf{Y}_n^k \right) \\
&= \frac{1}{2} \sum_n \left[ (Q_n^{p,c}(t))^2 + (P_n^c(t))^2 + (Q_i^{n,c}(t))^2 + (Z_e^{n,c}(t))^2 + (Y_n^{k,c})^2 \right].
\end{aligned}
\tag{16}
$$

Observing the state of all queues, we have the conditional expectation of on-step queuing evolutions, i.e., Lyapunov drift is:

$$
\Delta(\mathbf{\Theta}(t)) = \mathbb{E}\left[ L(\mathbf{\Theta}(t+1)) - L(\mathbf{\Theta}(t)) | \mathbf{\Theta}(t) \right].
\tag{17}
$$

Substituting the evolution expressions of all queues, we obtain the upper bound of Lyapunov drift is:

$$
\begin{aligned}
\Delta(\mathbf{\Theta}(t)) \leq B &- \sum_n \mathbb{E}\left[ Q_n^{p,c}(t) \left( R_n^{p,cs}(t) - N R_n^{k_n(t+1),cs} \right) \right] - \\
&\sum_n \sum_{i\neq n} \mathbb{E}\left[ Q_i^{n,c}(t) \left( \sum_{j|(j,i)\in L} \mu_{ji}^n(t) - \sum_{j|(i,j)\in L} \mu_{ij}^n(t) \right) | \mathbf{Q}^n(t) \right] \\
&- \sum_n \sum_e \mathbb{E}\left[ N R_n Z_e^{n,c}(t) \left( R_n^{k_n(t),cs} - R_n^{k_n(t+1),cs} \right) | \mathbf{Z}^n(t) \right] \\
&- \sum_k \mathbb{E}\left[ Y_n^{k,c}(R_n^{k,cs} - \gamma_n R_n^{k,cs}) | \mathbf{Y}_n^k \right].
\end{aligned}
\tag{18}
$$

Since the maximum transmission power is finite, all the rates of the network would be bounded. Additionally, the arrival rate $\lambda_n^{cmax}$ is also bounded. Hence, the parameter $B$ is a nonnegative constant.

Let the Lyapunov drift minus $V\mathbb{E}\left[\sum_{n,c} U(\bar{R}_n^{p,cs})|\Theta(t)\right]$, where $V$ is a weight parameter, we obtain the Lyapunov drift and penalty equation:

$$\Delta^U(\Theta(t)) \triangleq \Delta(\Theta(t)) - V\mathbb{E}\left[\sum_{n,c} U(\bar{R}_n^{p,cs})|\Theta(t)\right]. \tag{19}$$

According to the Lyapunov optimization theorem [22] and substituting (18) to (19), the upper bound of $\Delta^U(\Theta(t))$ can be expressed as:

$$
\begin{aligned}
\Delta^U(\Theta(t)) \leq B &- \sum_n \mathbb{E}\left[Q_n^{p,c}(t)\left(R_n^{p,cs}(t) - NR_n^{k_n(t+1),cs}\right)\right] - \\
&\sum_n \sum_{i \neq n} \mathbb{E}\left[Q_i^{n,c}(t)\left(\sum_{j|(j,i)\in L}\mu_{ji}^n(t) - \sum_{j|(i,j)\in L}\mu_{ij}^n(t)\right)|\mathbf{Q}^n(t)\right] \\
&- \sum_n \sum_e \mathbb{E}\left[NR_n Z_e^{n,c}(t)\left(R_n^{k_n(t),cs} - R_n^{k_n(t+1),cs}\right)|\mathbf{Z}^n(t)\right] \\
&- \sum_k \mathbb{E}\left[Y_n^{k,c}(R_n^{k,cs} - \gamma_n R_n^{k,cs})|\mathbf{Y}_n^k\right] - V\mathbb{E}\left[\sum_{n,c} U(\bar{R}_n^{p,cs})|\Theta(t)\right] \quad (RHS).
\end{aligned}
\tag{20}
$$

Rearranging and observing the *RHS* of (20), we find that the MSCAFB algorithm indeed minimizes the right hand side of (20).

If the arrival rates of each node are in the feasible region, based on the work in [22], there must exist a stationary scheduling, flow control and end-to-end encoding policy, which select the users and link rates independent of queue length and only relate to the channel statistics. This indicates that, if the channel statistics can be known a priori, the optimal control policy can be found as the solution of a deterministic policy. Let $U^*$ be the optimal value of problem (3), $R_n^{*cs}$ and $\mu_n^*$ denote the feasible and optimal arrival rate and transmission rate, respectively. Then, for all queues and any constants $\delta_1$, $\delta_2$ and $\delta_3$, there must exists a network control policy that is independent of all queue lengths and satisfies the following inequalities:

$$\sum_{\{i|(n,i)\in L\}} \bar{\mu}_{ni}^n \geq \mu_n^* + \delta_1, \tag{21}$$

$$\sum_{\{j|(i,j)\in L\}} \bar{\mu}_{ij}^n \geq \sum_{\{i|(j,i)\in L\}} \bar{\mu}_{ji}^n + \delta_2, \tag{22}$$

$$\bar{R}^e(t) \leq \bar{R}^{e*}(t) + \delta_3. \tag{23}$$

Since the MSCAFB indeed minimizes the RSH of (20), such that any stationary control policies (including the optimal policy) need to satisfy (20). Inserting (21)–(23) into (20), we get the following upper bound of our control algorithm:

$$RHS \, of \, (48) \leq B - \sum_n \delta_1\mathbb{E}[Q_n^{p,c}(t)] - \sum_n\sum_{i\neq n}\delta_2\mathbb{E}[Q_i^{n,c}(t)] - \sum_n\sum_e\delta_3\mathbb{E}[Z_e^{n,c}(t)] - VU^*. \tag{24}$$

Rearrange (24), we obtain the performance of MSCAFB algorithm.

## 4. Numerical Results and Discussions

For the network model presented in Figure 1, we consider i.i.d Rayleigh fading channels between nodes. The ratio of transmit power and noise has been normalized to 1. Let $h_{i,j}$ be the power gain between node $i$ and $j$, which follows exponential distribution and the mean of each link is presented in Table 2. The achievable rate between node $i$ and $j$ is $R_{i,j}(t) = \log(1 + h_{i,j}(t))$ and the rate of eavesdropper $R_{i,e}(t) = \log(1 + h_{i,e}(t))$. The utility function is a logarithmic utility function, i.e.,

$U_n^c(t) = \kappa + \log(R_n^{cs}(t))$, where $\kappa = 3$ and $R_n^{cs}(t)$ is the confidential rate selected by node $n$ in slot $t$. We assume the confidential data arrival process for each user follows an i.i.d Bernoulli process with rate $\lambda$.

**Table 2.** Mean Channel Gain.

| $(S_1,1)$ | $(S_1,2)$ | $(S_1,3)$ | $(1,d_1)$ | $(2,d_1)$ | $(3,d_1)$ |
|-----------|-----------|-----------|-----------|-----------|-----------|
| 6 | 8 | 10 | 8 | 6 | 4 |
| $(S_2,2)$ | $(S_2,3)$ | $(S_2,4)$ | $(2,d_2)$ | $(3,d_2)$ | $(4,d_2)$ |
| 6 | 8 | 10 | 8 | 6 | 4 |
| $(S_1,e)$ | $(S_2,e)$ | $(1,e)$ | $(2,e)$ | $(3,e)$ | $(4,e)$ |
| 3 | 2 | 1 | 1 | 2 | 3 |

Firstly, we analyze the performance of MSCAFB algorithm. In the simulation, the maximum average confidential data arrival rate is 2 bit/s, since the bandwidth is assumed to be one. Choosing the parameter $V \in \{10, 20, 80\}$, we get the results in Figures 3 and 4, where each value is collected by running 5000 times. Figure 3 shows the impact of increasing the average confidential data arrival rate on the utility function and Figure 4 depicts the average queue backlog in the network. From the simulation results, we find: (1) For a fixed $V$, when the arrival rate is low, Figure 3 indicates that the utility function linearly increases with the average admission confidential rate. The reason for this is that, if the arrival confidential rate is low, almost all the arrival confidential data can be admitted. (2) When the arrival confidential rate is larger than the secrecy channel capacity, the average admitted confidential rate turns into saturation. Not surprisingly, as the parameter $V$ increases, we observe that the utility function grows closer to the optimal value. (3) While in Figure 4, the average queue backlog is increased with $V$ dramatically. It indicates that the transmission delay is increased with $V$. Thus, the choice of $V$ is indeed a tradeoff between average utility and short-term system performance. To achieve both large utility and low delay, we will discuss the selection of parameter $V$.

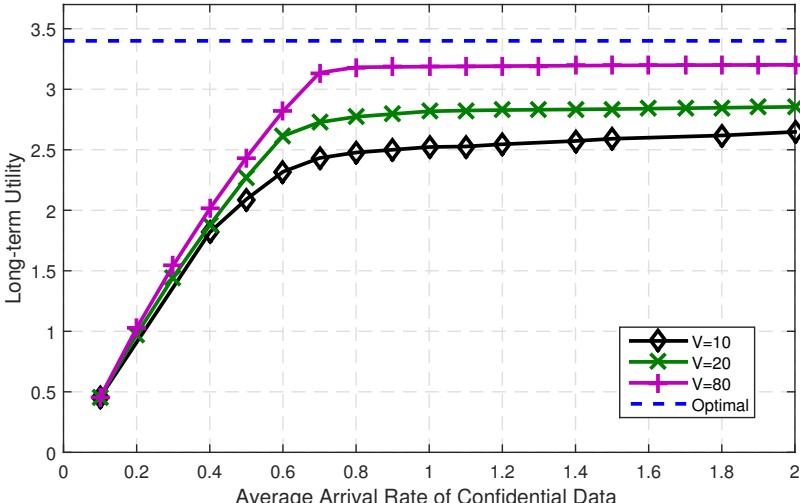

**Figure 3.** Long-term utility with varying confidential data arrival rate.

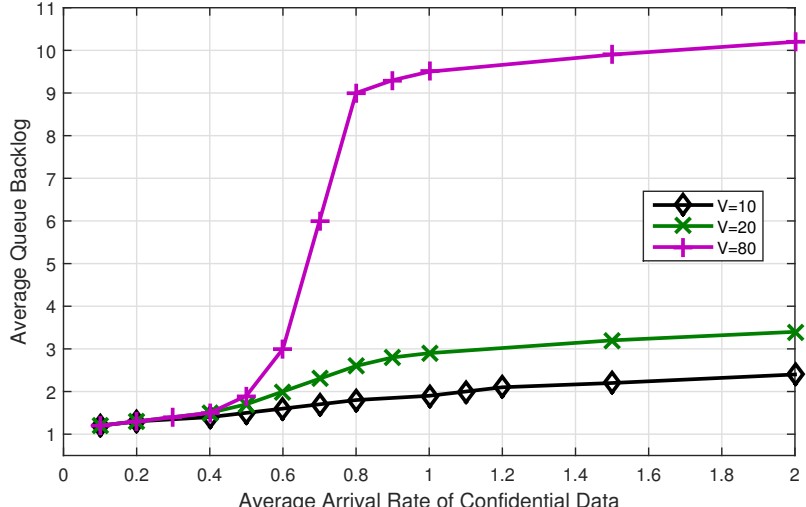

**Figure 4.** Average queue backlog with varying confidential data arrival rate.

According to [22], the long-term average utility is proportion to $1/V$, such that we can rewrite the utility function as $\sum_{n,c} U_n^c(\bar{r}_n^{cs}(t)) \approx \sum_{n,c} U_n^c(r_n^{*cs}) - BM/V$, where $\sum_{n,c} U_n^c(r_n^{*cs})$ denotes the optimal value and is a constant. Hence, the utility function is an increasing hyperbolic function of parameter $V$ and a good operating point would be to pick a $V$ value where an unit increases in $V$ yields a very small reduction in utility. At this point, the utility gains may not be worth the delay increase resulting from increasing $V$ (since delay is proportional to $V$). Let $\eta > 0$ be the slope of utility function $\sum_{n,c} U_n^c(\bar{r}_n^{cs}(t))$. Differentiating to $V$, i.e., $d\left(\sum_{n,c} U_n^c(r_n^{*cs}) - BM/V\right)/dV = BM/V^2 = \eta$, we have the good operating point of $V = \sqrt{BM}$, where $M$ is the number of nodes, and $B \triangleq \frac{1}{M}\sum_{n=1}^{M}\left[\left(\lambda_n^{cmax} + R_{n,max}^{in,s}\right)^2 + (R_{n,max}^{out,s})^2\right]$. Based on the setting of simulation, i.e., $M = 8$, $B \approx 20$ and $\eta \approx 0.5$, we can obtain the good operating point $V \approx 20$.

Next, we analyze the performance of MSCAFB algorithm and compare with MSCA algorithm in [23]. Figure 5 reflects the influence of secrecy codeword length on the network utility, as well as comparing with infinite secrecy codeword. The average arrival of confidential message is 2 bit/s and the parameter $V$ is 80. The maximum allowable portion of confidential message $\gamma_n$ is chosen from $\{0.05, 0.1, 0.2\}$, and the secrecy codeword length varies from $\{50, 100, 500, 1000, 2000, 4000\}$ bits. From Figure 5, we find: (1) When the secrecy codeword length is 50 bits, the network utility is only 30% of the optimal value. (2) With the increasing of secrecy codeword length, the network utility is increased. When secrecy codeword length is up to 1000 bits, the network utility trends to be gentle, but closes to the value with infinite secrecy codeword. The reason for this is that, when the secrecy codeword length is small, subject to the constraint of $\gamma_n$, the confidential message inserted to the codeword is decreased. Such that the network utility is low, and vice versa. (3) The network utility is increased with the value of $\gamma_n$. The reason is that, with $\gamma_n$ increasing, more confidential message can be inserted into a secrecy codeword. Particularly, when $\gamma_n = 0.2$ and secrecy codeword length is 1000 bits, the network utility is larger than that of infinite secrecy codeword. In order to depict the influence of $\gamma_n$ on the network utility, we have Figure 6.

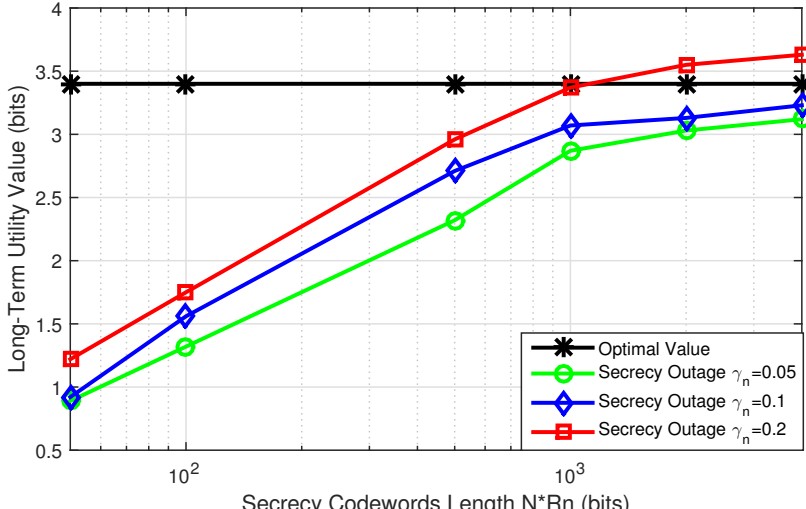

**Figure 5.** Long-term utility with varying secrecy codeword length.

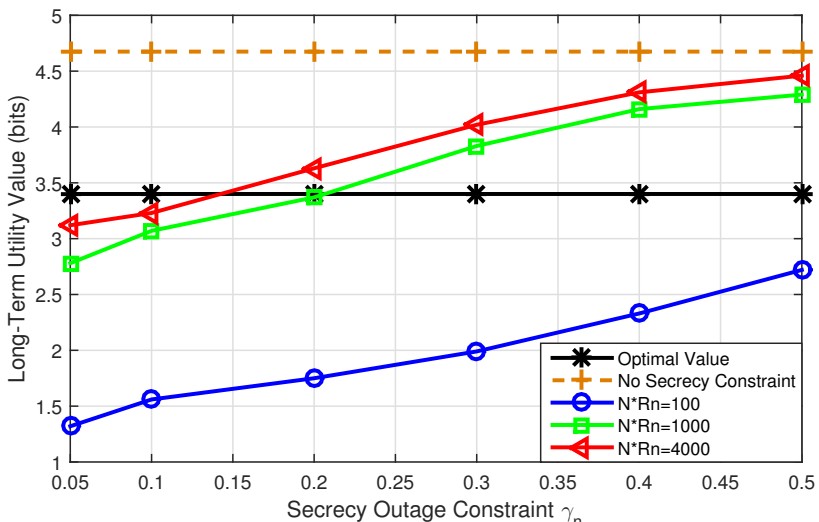

**Figure 6.** Long-term utility with varying secrecy outage constraint.

In Figure 6, assuming the secrecy codeword length is $\{100, 1000, 4000\}$ bits, we vary $\gamma_n$ from 0.05 to 0.5. The results show that: (1) With $\gamma_n$ increasing, we have the similar results with Figure 5. (2) With the increasing of secrecy codeword length, the threshold of $\gamma_n$ at which network utility is larger than that of infinite secrecy codeword is decreased. The reason is that larger secrecy codeword length means more confidential message can be inserted into a secrecy codeword. Particularly, when the secrecy codeword length is small, even $\gamma_n$ is large, the network utility can not exceed the optimal value. (3) Although both $\gamma_n$ and secrecy codeword length are large, the network utility can exceed the optimal value, it can not exceed the network capacity without secrecy constraint.

Each node needs the information of queue length from eavesdroppers to decide the control algorithm, considers a situation that queue length information is shared among all the nodes, but in some environment it is impossible to know this information precisely. Compared with getting perfect information about $Z_e^{n,c}(t)$, it is more realistic to know the time-average packet arrival rate of eavesdroppers. Considering this, we propose an imperfect estimation of $Z_e^{n,c}(t)$. The estimated queue length in legitimate node is:

$$\hat{Z}_e^{n,c}(t+1) = [\hat{Z}_e^{n,c}(t) - \bar{R}^e(t)]^+ + (R_n^{k,cs} + \iota) \tag{25}$$

where $\iota$ is an over-estimated slack variable to queue stability. As to the control algorithm, we use $\hat{Z}_e^{n,c}$ to substitute $Z_e^{n,c}$ in resource allocation algorithm.

## 5. Conclusions

In this paper, we consider the online control problem of a multi-hop wireless network with a security constraint. To guarantee confidentiality in multi-hop transmission, we employed an independent randomization encoding strategy with infinite and finite secrecy codewords. Using the stochastic network optimization, we develop a dynamic control algorithm for finite secrecy encoding strategies. We also proved that the proposed control algorithm achieve an utility close to the optimal value asymptotically. Finally, we simulate the online control algorithm with various network scenarios. The results demonstrate that the value of utility approaches the optimum, while the average queue backlog increases very fast. Thus, how to make a tradeoff between performance and queue backlog should be the subject of future research.

**Author Contributions:** Q.L. proposed the idea, derived the results and wrote the paper. S.L. reviewed the article in initial and revised versions. C.Z. assisted to revise the paper. X.Q. and H.X. assisted in revising the paper.

**Funding:** This work was supported in part by National Natural Science Foundation of China (No. 61761021), Natural Science Foundation of Jiangxi Province (Grant No. 20181bab202018), Projects of Humanities and Social Sciences of universities in Jiangxi (JC18224, JY161012) and the Doctoral Research Fund of Jiangxi University of Science and Technology.

**Conflicts of Interest:** The authors declare no conflict of interest.

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
