# Peer review of "Secrecy Control of Wireless Networks with Finite Encoding Blocklength"

_algorithms, doi:10.3390/a12020049_

Round 1

Reviewer 1 Report

System model is not fully justified and it requires improvement and clear goals of the model to be
highlighted.

Author Response

First of all, we would like to express our gratitude to the anonymous reviewers for his/her helpful suggestions and valuable comments. We have read the reviewers' comments carefully and revised the paper accordingly. We will address specific comments the reviewers have made and provide our response individually. For the convenience of the reviewers, we identify reviewer's every question/suggestion in bold font with blue color, followed by our response and the corresponding remark on the modifications of the paper with bold font.

Reviewer 2 Report

In the first section authors provide a wide scope of related works, but most of the publications are mentioned in one sentence and reader is directly linked to the article. It is strongly advised to extend tis section, especially present the content of pointed publications - not only the topic.

The proposed algorithm is explained in a very hard way. Authors provide many formulas that are are not precisely explained. Figure 2 is unclear (wrongly exported) and the reader can not see key elements that are explained. Also the caption of the first figure is not correct. It is strongly advised to review the algorithm description, it should be explained also by words, not only by the formulas and variable description.

In the reviewer point of view the key assumption of the proposed algorithm is not correct - "Source node n needs to track the accumulated of confidential message at each eavesdropper". In a real network any node can check if somebody has intercepted the transmission it is just impossible to estimate that. We can only suspect that somebody is catching the transmission, the probability that specific information was captured is unknown.

Another issue of the article is that the relation of the proposed algorithm to sensor networks is not well enough described. Authors point the structure of the network, but the information about the exemplary network are not clear enough. The numerical results presented in the 4th section looks good, but their assumptions are rather general, not referred to a real network.

From technical point of view the article has some issues. The language in many paragraphs is not correct. The style of the sentences causes that the merit content may not be clear for the reader. In the text also many mistakes can be found. In the reviewer point of view an extensive editing of the English language and style is required.

Author Response

(The authors gave the same response as above.)

Round 2

Reviewer 2 Report

Current form of the article contains all significant and necessary improvements. In the response letter Authors provide an explanation of introduced changes. 

In the introduction the description of related works was expanded by introducing a wider description of the key elements of other studies.

Authors clarified the disruption of proposed algorithm, also the key assumption was also explained. The current form provides more important information for the reader. 

From technical point of view the article was significantly improved. in current state it is advised to only check the language, in some cases Authors can find a new ideal for some sentences.